# Assessment of low-vault cases with an implantable collamer lens

**Sayaka Kato**(ID)*, **Kimiya Shimizu**◊, **Akihito Igarashi**◊

Eye Center, Sanno Hospital, Tokyo, Japan

◊ These authors contributed equally to this work.
* sayaka331010843@yahoo.co.jp

## Abstract

This study aimed to examine clinical results in low-vault eyes after implementation of a Hole implantable collamer lens (KS-AquaPORT™, STAAR Surgical Company) in terms of visual outcomes and complications over a one-year follow-up period. This was a retrospective cohort study of subjects who underwent Hole implantable collamer lens surgery at Sanno Hospital, exhibited low vault, and were followed up for 1 year. Patients were included if they met the following criteria: 20≤ age ≤55 years; stable refraction ≥6 months; -1.0 to -20.0 diopters of myopia; endothelial cell density ≥1800 cells/mm$^2$; and no history of ocular surgery, progressive corneal degeneration, cataract, glaucoma, or uveitis. Main outcome measurements were the safety and efficacy indices, predictability, and vault. Values were indicated as the mean ± standard deviation. Subjects included 16 patients (age: 38 ± 8 years; 6 males; 25 eyes). Toric lenses were utilized for 10 eyes. Implantable collamer lens size was 12.1, 12.6, and 13.2 mm for 18, 6, and 1 eye(s), respectively. One year postoperatively, the safety index was 1.07; for 22 eyes with a target refraction of that of emmetropic eyes, the efficacy index was 0.90; and 96% of eyes were within ± 0.50 diopters of attempted versus achieved spherical equivalent correction. Postoperative vault was 142 ± 60 μm. One year postoperatively, no additional surgery was required for rotation of toric implantable collamer lens, and no advanced cataracts, increased intraocular pressure, or decreased endothelial cells were observed. In conclusion, Hole implantable collamer lens yielded satisfactory visual outcomes and no postoperative complications for low-vault eyes, suggesting its suitability for such cases.

## Introduction

The implantable collamer lens (ICL) is generally the first choice for surgical correction of high myopia [1]. However, the number of ICLs implanted to treat low-to-moderate myopia is increasing [2–5]. The posterior chamber phakic intraocular lens with a central hole (Hole ICL; KS-AquaPORT™; STAAR Surgical Company, Monrovia, CA, USA), invented by Shimizu et al. [6], has a central artificial hole with a 0.36-mm diameter, and was approved for use in Japan by the Ministry of Health, Labour and Welfare in 2014. The Hole ICL has been reported as safe and effective for correcting refractive errors, and it provides predictable, stable, and

**Competing interests:** The authors have declared that no competing interests exist.

satisfactory visual outcomes [6, 7]. The introduction of the Hole ICL (V4c) was a marked improvement over the Visian ICL V4 (STAAR Surgical Company), which has been associated with cataract formation and increased intraocular pressure (IOP) caused by postoperative complications. The central hole improves aqueous humor circulation, preventing many of these complications, without the need for peripheral iridectomy, as is required when utilizing the ICL V4 [6, 8, 9].

An appropriate vault (the distance between the posterior ICL surface and the anterior crystalline lens surface) following ICL implantation is generally 250–750 μm, or 0.5–1.5 times the corneal thickness. Patients with a low vault before ICL V4 implantation are at an increased risk of postoperative cataract progression and toric ICL rotation [10, 11]. Shimizu et al. [8], utilizing Hole ICL, reported a postoperative cataract rate of 0% over a 5-year follow-up; for three studies, 0.3–10% of eyes required additional surgery due to toric ICL rotation [7, 12, 13], although not all of these eyes exhibited a low vault. To our knowledge, no postoperative results of specifically low vault cases (<250 μm) after Hole ICL have been reported. Therefore, in the current study, we specifically examined the clinical results of such cases.

## Materials and methods

This single-center, retrospective cohort study was approved by the institutional review board of Sanno Hospital, Tokyo, Japan (International University of Health and Welfare; 16-S-24) and followed the tenets of the Declaration of Helsinki. The requirement for informed consent was waived in view of the retrospective design of the study and the anonymity of the data.

### Study population

The subjects were 16 patients (age: 38 ± 8 years; 6 males and 10 females; 25 operated eyes) who underwent Hole ICL surgery at the Sanno Hospital Eye Center, exhibited low vault (<250 μm), and were followed up from the first day postoperatively for 1 year. Preoperative data are summarized in Table 1. Patients were considered candidates for this surgical technique if they met the following criteria: 20≤ age ≤55 years; stable refraction for at least 6 months; -1.0 to -20.0 diopters (D) of myopia; endothelial cell density (ECD) ≥1800 cells/mm$^2$; and no history of ocular surgery, progressive corneal degeneration, cataract, glaucoma, or

**Table 1. Preoperative data of the study population.**

| | |
|---|---|
| N (eyes, subjects) | 25, 16 |
| Males | 6 (37.5%) |
| Age (years) | 38 ± 8 (23 to 54) |
| Spherical equivalent (D) | -7.40 ± 4.06 (-1.00 to -15.50) |
| Subjective astigmatism (D) | -1.43 ± 1.92 (0.00 to -8.00) |
| Horizontal angle-to-angle (mm) | 11.77 ± 0.43 (11.00 to 12.79) |
| Vertical angle-to-angle (mm) | 11.99 ± 0.47 (11.20 to 12.98) |
| Keratometric values (D) | 43.78 ± 1.85 (37.88 to 46.63) |
| Axial length (mm) | 26.30 ± 1.76 (22.09 to 28.96) |
| ICL size (mm) | 12.1; 18 eyes (72%) |
| | 12.6; 6 eyes (24%) |
| | 13.2; 1 eye (4%) |
| Eyes with toric ICL implantation | 10 eyes (40%) |

Data are presented as number, number (percentage), or mean ± standard deviation (range), as applicable.

D = diopter; ICL = implantable collamer lens.

uveitis. The uncorrected distance visual acuity (UDVA), corrected distance visual acuity (CDVA), and subjective refraction spherical equivalent were measured before surgery and at 1 day, 1 week, 1 month, 3 months, and 1 year postoperatively. Keratometric values and IOP were measured using an autokeratometer/autotonometer (TONOREF™ III; NIDEK Co., Ltd., Gamagori, Japan). Corneal ECD was measured using a noncontact specular microscope (FA-3809IIP, Konan Medical, Inc., Nishinomiya, Japan). Axial length was measured using an optical biometer (IOLMaster 700; Carl Zeiss Co., Ltd., Kojimachi, Japan). Preoperative measurements of anterior chamber depth (ACD) and angle-to-angle (ATA), as well as postoperative measurements of the central vault, were obtained using swept-source, anterior segment, optical coherence tomography (CASIA2; Tomey Corporation, Nagoya, Japan) under photopic conditions (background illumination = 400 lx).

Patient data were anonymized before access and/or analysis. Written informed consent for the surgery was obtained from all patients after explanation of the nature and possible consequences of the surgery.

## Implantable collamer lens power and size calculation

The postoperative target refractive power of the ICL was that of an emmetropic eye. However, for people over 40 years old, presbyopia was taken into consideration; for these patients' eyes, a simulation was performed using soft contact lenses, and mild myopia was the target refractive power, according to the wish of the patients. The appropriate size of the ICL was calculated by measuring the ATA and using the formula reported by Igarashi et al. [14].

## Surgical procedure

Patients did not undergo iridotomy or intraoperative peripheral iridectomy before implantation of a Hole ICL; however, they did receive dilating and cycloplegic agents on the day of surgery. Under topical anesthesia, a viscosurgical device (Opegan®; Santen Pharmaceutical Co., Ltd., Osaka, Japan) was inserted into the anterior chamber, and a Hole ICL was inserted through a 3-mm clear corneal incision using an injector cartridge (STAAR Surgical Company). The ICL was placed in the posterior chamber, after which the viscosurgical device was washed out of the anterior chamber using a balanced salt solution. Thereafter, a miotic agent (acetylcholine chloride; Ovisort; Daiichi Sankyo Company, Limited, Tokyo, Japan) was instilled. In order to suppress potential cyclotorsion in the supine position during the process of toric ICL implantation, the horizontal axis was marked preoperatively by means of a slit-lamp biomicroscope. The ICL was rotated by 15.0° or less using a manipulator. Its position was fixed either horizontally or vertically to optimize the postoperative vault. All surgeries were performed by two experienced surgeons (KS and AI). Postoperatively, 0.1% betamethasone (Rinderon®; Shionogi Pharma Co., Ltd., Osaka, Japan) and 0.5% levofloxacin (Cravit®; Santen Pharmaceutical Co., Ltd.) were administered topically, four times daily for 1 week, with gradual reduction of the dose thereafter.

## Patient and public involvement

No patients or members of the public were involved in setting up the research question or developing the outcome measures, nor were they involved in developing plans for the design or implementation of this study. No patients or members of the public were consulted for advice on interpretation or writing up of the results, nor was the burden of the interventions on patients assessed. The results of the research were to be disseminated to participants upon request.

## Statistical analysis

Statistical analysis was conducted using commercially available statistical software (Ekuseru-Toukei 2015, Social Survey Research Information Co., Ltd., Tokyo, Japan). The normality of all data samples was first evaluated using the Kolmogorov-Smirnov test. As the use of parametric statistics was not appropriate, the Friedman repeated measures analysis of variance (ANOVA) was used for the analysis of changes over time. Unless otherwise indicated, results are expressed as the mean ± standard deviation, and a $P$-value $< .05$ was considered statistically significant.

## Results

### Study population

Preoperative patient demographics are summarized in Table 1. Logarithm of the minimum angle of resolution (logMAR) UDVA and CDVA were 1.33 (Snellen equivalent = 20/400) ± 0.42 (range, 0.40 to 2.00) and -0.15 (Snellen equivalent = 20/16) ± 0.10 (range, -0.30 to 0.00), respectively. ACD was 2.95 ± 0.25 mm (range, 2.47 to 3.35 mm). IOP was 12.8 ± 2.6 mmHg (range, 9.0 to 18.0 mmHg). ECD was 2,711 ± 275 cells/mm$^2$ (range, 2,288 to 3,236 cells/mm$^2$). The horizontal ATA was 11.77 ± 0.43 mm (range, 11.00 to 12.79 mm). The vertical ATA was 11.99 ± 0.47 mm (range, 11.20 to 12.98 mm). The fraction of toric ICLs was 40% (10/25 eyes).

### Safety outcomes

At 1 year after surgery, the logMAR CDVA was -0.18 (Snellen equivalent = 20/16) ± 0.11 (range, -0.30 to 0.00) and the safety index (mean postoperative CDVA/mean preoperative CDVA) was 1.07 ± 0.20. One year postoperatively, 18 eyes (72%) exhibited no change in CDVA, 4 eyes (16%) gained one line, 1 eye (4%) gained two lines, and 2 eyes (8%) lost one line (Fig 1).

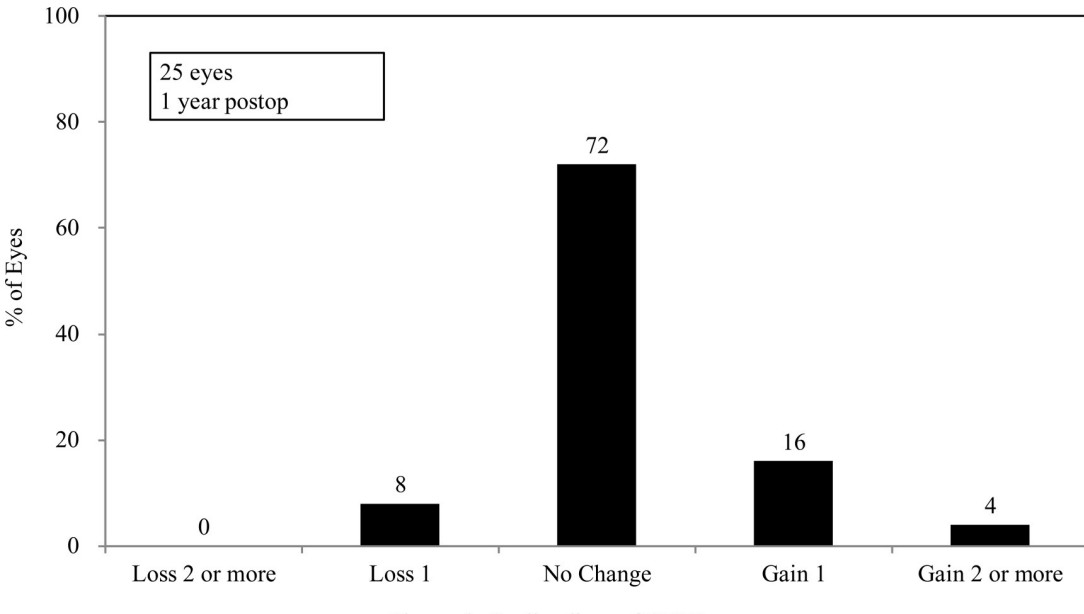

**Fig 1. Changes in CDVA after implantation of an implantable collamer lens with a central hole.** CDVA = corrected distance visual acuity.

### Effectiveness outcomes

For 22 eyes, the target refractive power was that of an emmetropic eye. One year after surgery, the logMAR UDVA for these eyes was -0.10 (Snellen equivalent = 20/16) ± 0.10 (range, -0.30 to 0.10), the efficacy index (mean postoperative UDVA/mean preoperative CDVA) was 0.90 ± 0.20, and all cases had an UDVA of 20/25 or better (Fig 2).

### Predictability

Fig 3 is a scatter plot of attempted versus achieved spherical equivalent correction. One year after surgery, 96% and 100% of eyes, respectively, were within ±0.5 D, ±1.0 D of the attempted correction.

### Stability

Fig 4 illustrates the change over time in manifest refraction of 22 eyes with a target refraction of that of an emmetropic eye. The change in manifest refraction from 1 day to 1 year postoperatively was -0.13 ± 0.26 D (range, -0.75 to 0.25 D). There was no statistically significant change in manifest refraction over time (ANOVA, p = 0.290).

### Vault

Fig 5 depicts the change over time in vault. The change in the vault from 1 day to 1 year postoperatively was -42 ± 55 μm (range, -183 to 52 μm). There was no statistically significant change in vault over time (ANOVA, p = 0.210).

### Intraocular pressure

Fig 6 demonstrates the change over time in IOP. The change in IOP was 0.3 ± 1.9 mmHg (range, -5.0 to 3.0 mmHg) from before surgery to 1 year postoperatively. There was no statistically significant change in IOP (ANOVA, p = 0.782).

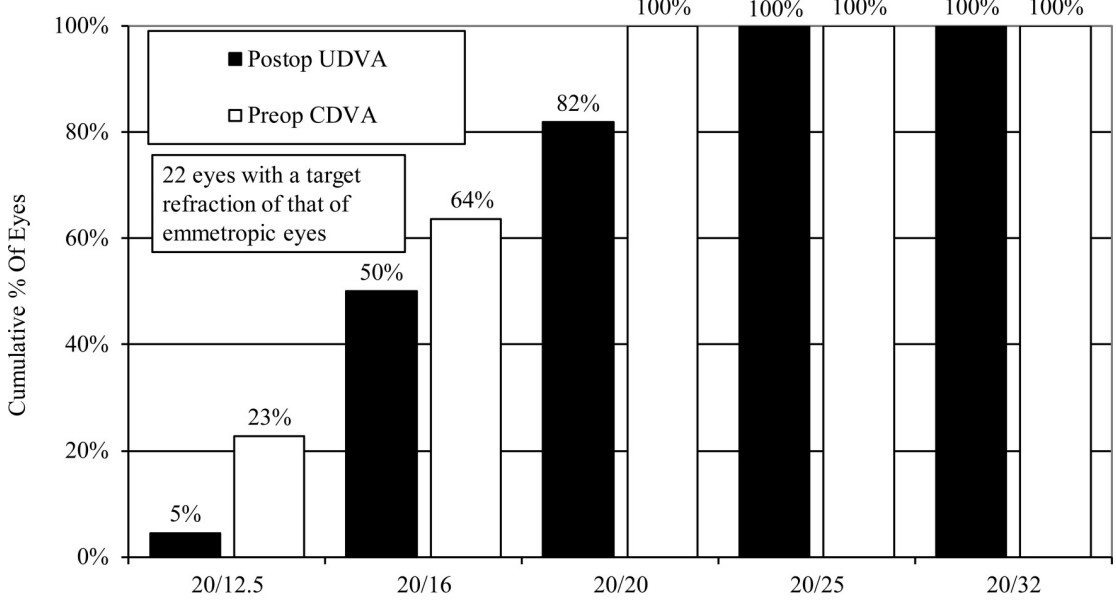

**Fig 2. Changes in UDVA after implantation of an implantable collamer lens with a central hole.** CDVA = corrected distance visual acuity, UDVA = uncorrected distance visual acuity.

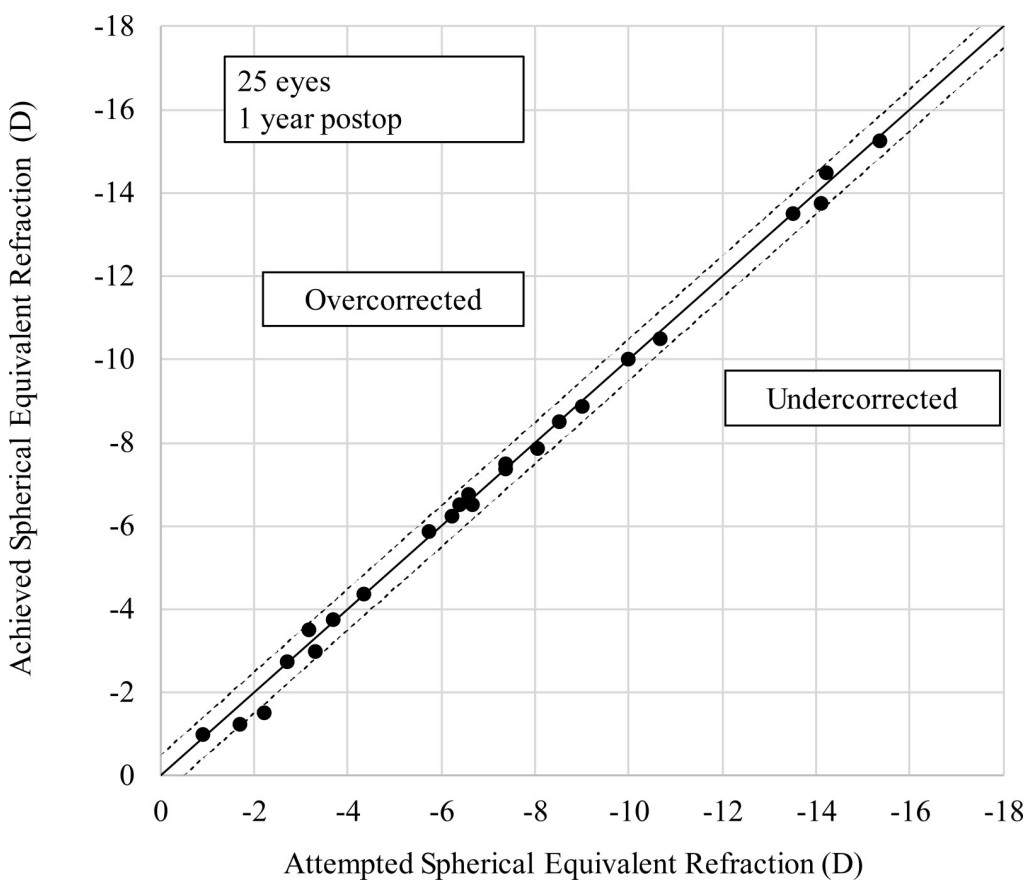

**Fig 3. A scatter plot of attempted versus achieved correction (spherical equivalent) after implantation of an implantable collamer lens with a central hole.** D = diopter.

## Endothelial cell density

The preoperative, 3-month, and 12-month postoperative ECD was $2,711 \pm 275$ cells/mm$^2$, $2,687 \pm 261$ cells/mm$^2$, and $2,751 \pm 272$ cells/mm$^2$, respectively. There was no statistically significant change in ECD (ANOVA, $p = 0.666$). The mean percentage change in ECD was $-1.7 \pm 5.5\%$ (-10.4% to 6.1%) from before surgery to 1 year postoperatively.

## Complications

There were no intraoperative or postoperative complications, and all implantation procedures were uneventful.

## Discussion

In this study, we observed 25 eyes with low vault (<250 μm) for 1 year; there were no complications, including postoperative cataracts, and the clinical results were satisfactory.

Kojima et al. [15] reported that, one year postoperatively, 13.9% of patients had a low vault (≤250 μm), 72.2% had a moderate vault (250–750 μm), and 13.9% had a high vault (≥750 μm). Myopic eyes with ICLs are thicker at the periphery than in the center. However, currently, it is challenging to measure the peripheral vault using anterior segment OCT, as the iris interferes with the measurement. The addition of a hole in the ICL allows the circulation of

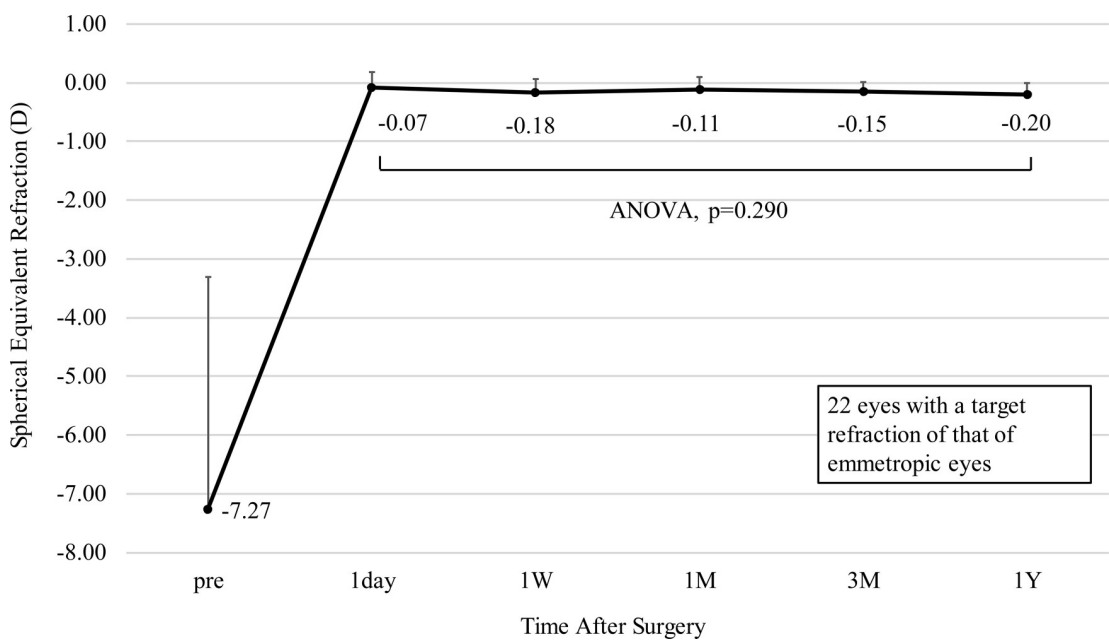

**Fig 4. Spherical equivalent refraction after implantation of an implantable collamer lens with a central hole.** The points depict mean values and the error bars depict the upper bound of the standard deviation. ANOVA = analysis of variance, D = diopter, W = week, M = month(s), Y = year.

the aqueous humor and reduces complications, even in eyes with a low vault. Therefore, vault evaluation in this study was performed only for the center of the lens. Postoperative vault may change over time, and it has been reported that eyes receiving a conventional ICL exhibited a

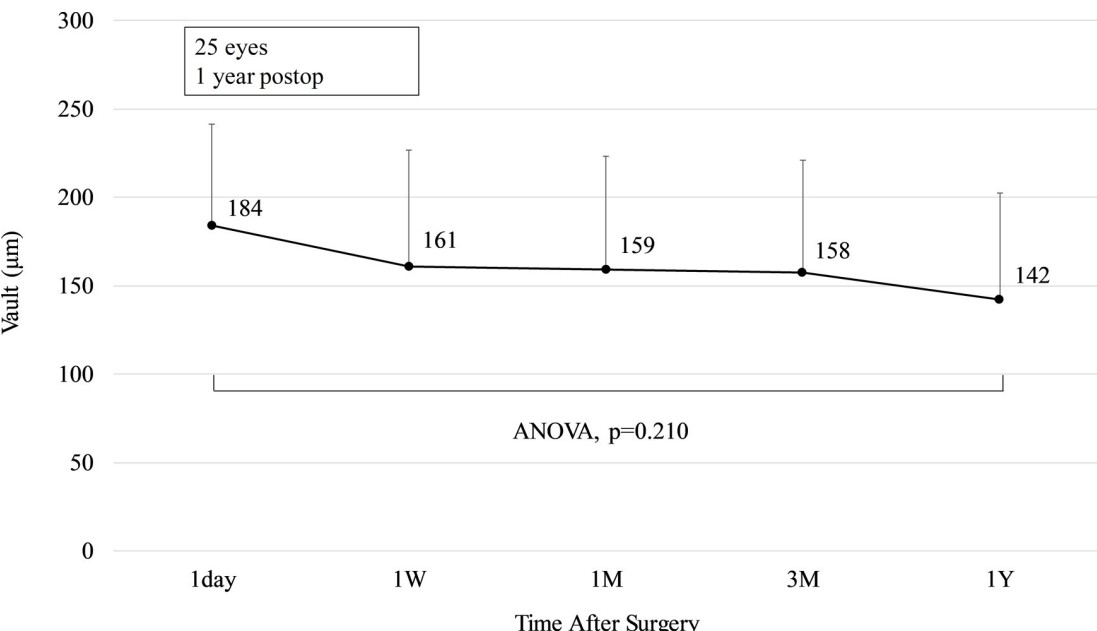

**Fig 5. Change in vault after implantation of an implantable collamer lens with a central hole.** The points depict mean values and the error bars depict the upper bound of the standard deviation. ANOVA = analysis of variance, W = week, M = month(s), Y = year.

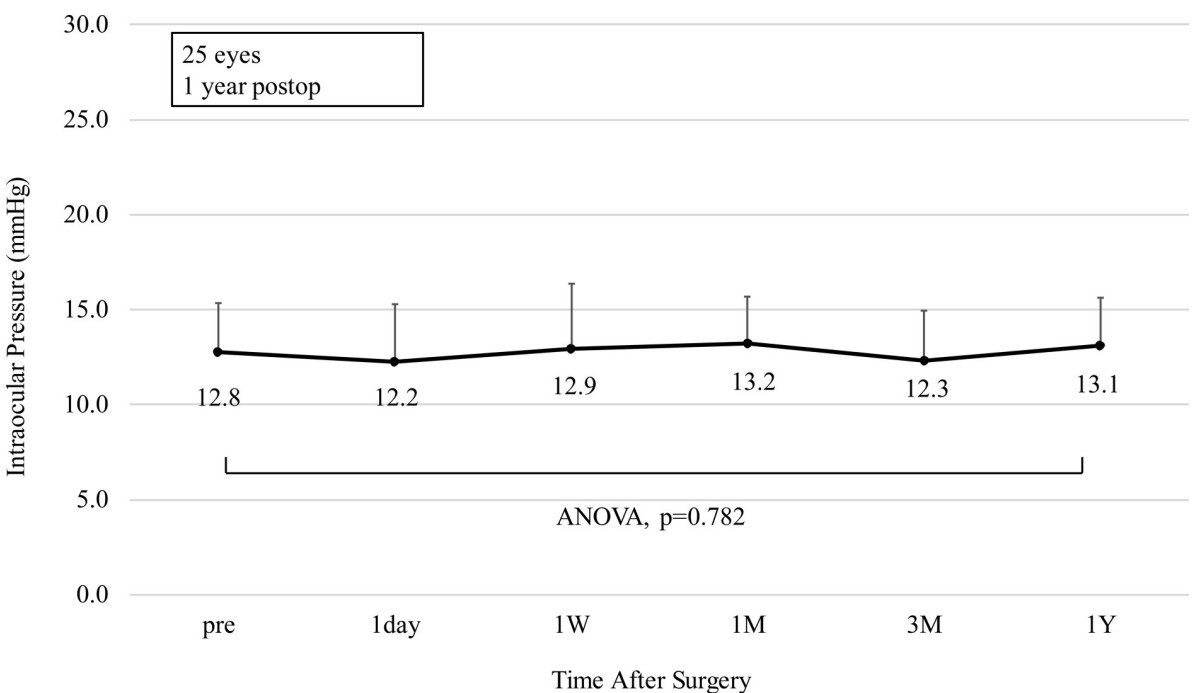

**Fig 6. Change in intraocular pressure after implantation of an implantable collamer lens with a central hole.** The points depict mean values and the error bars depict the upper bound of the standard deviation. ANOVA = analysis of variance, W = week, M = month(s), Y = year.

reduced vault, from 466 ± 218 μm directly after surgery, to 184 ± 159 μm, 10 years later [10]. In addition, Kojima et al. [15] examined vault change separately for patients with postoperative high vault (>750 μm), moderate vault (250–750 μm), and low vault (<250 μm). They reported that high vault decreased statistically significantly up to 3 months after surgery, moderate vault decreased statistically significantly up to1 week after surgery, and low vault remained unchanged from 1 day to 1 year after surgery. Similarly, no statistically significant changes were observed in the present study, as all eyes under investigation exhibited low vault. It has been reported that vault statistically significantly decreased from 589.7 μm, 1 month after Hole ICL surgery, to 458.3 μm, 2 years after surgery [16]. Another report revealed no statistically significant change in vault from one day after surgery (426 μm) to two years after surgery (449 μm) [17]. However, these results were not stratified by initial vault size. The current study has a follow-up time of one year after surgery, and lens thickness was not measured; however, a long-term reduction in vault is related to lens thickness, which changes with age [10]. It is generally reported that the crystalline lens bulges 20 μm/year anteriorly [18], and the ACD has been reported to decrease 11 μm/year due to aging of the crystalline lens [19]. These may decrease vault over the long term. Therefore, an extended follow-up time is required, in future studies.

Postoperative complications due to low vault include cataract progression and an astigmatic axis shift due to rotation of the toric ICL. In previous studies, anterior subcapsular opacities occurred after conventional ICL in 1.1–5.9% of cases by 5 years after the operation, and 0–1.8% of total cases required cataract surgery [8, 20–22]. Those cases requiring surgery all had a mean vault of <270 μm. Conversely, none of the cases receiving Hole ICL surgery exhibited anterior subcapsular opacities or visually significant cataracts, irrespective of vault [8]; this is likely due to the improvement in aqueous humor circulation of Hole ICL compared with

conventional ICL. Goukon et al. [23] reported that the mean central ECD loss was 0.3%, 2 years after Hole ICL. Although no statistically significant decrease in ECD was observed in our study, the sample size was small and the follow-up was only 1 year after surgery; future studies with long-term follow-up periods are required.

Sheng et al. [11] reported that a lower vault was correlated with an increased risk of Toric ICL rotation. On the other hand, Lee et al. [24] examined the factors leading to toric ICL rotation of Hole ICL and discovered no statistically significant correlation with vault. In the present study, the sample size for toric ICL was small (10 eyes); nonetheless, no rotation was observed in these low vault eyes. We believe that the relationship between low vault of eyes receiving a Hole ICL and rotation of toric ICL warrants further investigation, with larger sample sizes.

Overall, these results suggest that, for low vault eyes, Hole ICL yields satisfactory results for at least the first year after surgery, with no postoperative complications. Further research is required, with a longer follow-up time and larger sample size.

## Supporting information

**S1 Data.**
(XLSX)

## Author Contributions

**Conceptualization:** Kimiya Shimizu.

**Formal analysis:** Sayaka Kato, Akihito Igarashi.

**Funding acquisition:** Kimiya Shimizu.

**Investigation:** Sayaka Kato, Akihito Igarashi.

**Methodology:** Sayaka Kato, Akihito Igarashi.

**Project administration:** Kimiya Shimizu.

**Supervision:** Kimiya Shimizu.

**Validation:** Sayaka Kato.

**Writing – original draft:** Sayaka Kato.

**Writing – review & editing:** Kimiya Shimizu, Akihito Igarashi.

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
