## [Decision Letter · Decision Letter 0]

24 Sep 2020

PONE-D-20-24129

Assessment of low-vault cases with an implantable collamer lens

PLOS ONE

Dear Dr. Kato,

Thank you for submitting your manuscript to PLOS ONE. After careful consideration, we feel that it has merit but does not fully meet PLOS ONE’s publication criteria as it currently stands. Therefore, we invite you to submit a revised version of the manuscript that addresses the points raised during the review process.

We look forward to receiving your revised manuscript.

Kind regards,

Hidenaga Kobashi, M.D., Ph.D.

Academic Editor

PLOS ONE

Journal Requirements:

Additional Editor Comments (if provided):

The reviewers and I have completed their assessments of your manuscript, Assessment of low-vault cases with an implantable collamer lens (PONE-D-20-24129) and would like to publish it in the journal once you have responded to the referees' comments (enclosed below). In the cover letter with the revised manuscript, please indicate how each of the reviewers' suggestions was addressed.

Reviewers' comments:

Reviewer's Responses to Questions

**Comments to the Author**

1. Is the manuscript technically sound, and do the data support the conclusions?

Reviewer #1: Yes

Reviewer #2: Yes

2. Has the statistical analysis been performed appropriately and rigorously? 

Reviewer #1: Yes

Reviewer #2: Yes

3. Have the authors made all data underlying the findings in their manuscript fully available?

Reviewer #1: Yes

Reviewer #2: Yes

4. Is the manuscript presented in an intelligible fashion and written in standard English?

Reviewer #1: Yes

Reviewer #2: Yes

5. Review Comments to the Author

Reviewer #1: The authors focused on low vault cases after implantable collamer lens implantation surgery and evaluated the efficacy, safety, predictability, and complications.

It has been pointed out that the low vault has a risk of cataract formation and toric ICL rotation. For this reason, it is important to collect and analyze only low vault cases, which seems to be the novelty of this paper.

1. In the Introduction and Discussion section, the low vault is referred to as a case of less than 250 µm. However, the Methods section doesn't show at what point in time or how many vaults were defined as low vaults.

2. Citing Lee's paper in the Discussion section, the authors state that the vault is not related to the toric ICL rotation, but some studies have reported that the vault is correlated (Sheng XL, Rong WN, Jia Q, et al. Outcomes and possible risk factors associated with axis alignment and rotational stability after implantation of the Toric implantable collamer lens for high myopic astigmatism. Int J Ophthalmol. 2012;5(4):459-465.). This issue should be mentioned in detail in the discussion section.

3. Authors measure the central vault, but in myopic ICL, the peripheral vault is closest to the lens due to the shape of the lens. The limitations of measuring only the central vault should be mentioned in the discussion.

4. It has been reported that vault measurements dynamically change in examination environment such as brightness. CASIA measurement conditions should be added in Methods section.

Reviewer #2: Authors specifically examined the cases with low vault after Hole ICL, then reported the safety and efficacy of Hole ICL only just for one year. This research is clinically relevant, but I actually don't know if the sample volume that authors examined in this clinical research is sufficient to come to the conclusion. Also, authors observed the cases just for 1 year. The safety of the surgery would not be able to be seen in one year only.

1. Authors mentioned that the aim in this research was to examine the clinical results of visual outcomes and long-term complications. But just 1-year follow-up is not long term. Authors should change the purpose of this research.

2. Authors examined 16 patients, 25 cases who underwent Hole ICL surgery, then exhibited low vault. How frequently have the authors seen the cases with low vault after Hole ICL surgery ? Readers may not understand how commonly it is observed after Hole ICL. Authors should address the background information on it in either Introduction or Discussion section.

3. Abstract, Line 23 “implantable collamer lens size was 12.1, 12.6, and 13.2 mm for…”

Please create a large capital letter at the beginning of a paragraph.

4. Citation 8, This is a meta-analysis and review paper. Authors should cite the original paper.

5. ICL is a posterior chamber lens, but author should have a discussion regarding postoperative ECD in Discussion. ECD loss is one of the most serious complications.

6. PLOS authors have the option to publish the peer review history of their article (what does this mean?). If published, this will include your full peer review and any attached files.

Reviewer #1: No

Reviewer #2: No

---

## [Author Response · Author response to Decision Letter 0]

19 Oct 2020

In this study, we set out to investigate the clinical results of low-vault eyes implanted with an implantable collamer lens with a central hole (Hole ICL). It has previously been demonstrated that the use of Hole ICL may reduce postoperative complications such as cataract formation and an increased intraocular pressure, most likely due to an improved aqueous humor circulation provided by the hole. Moreover, patients with low vault (<250 μm) following conventional ICL implantation are at an increased risk of postoperative cataract progression and toric ICL rotation. We believe that our study makes a significant contribution to the literature because we have demonstrated the safety, efficacy, and predictability of Hole ICL implantation specifically in low-vault eyes over a 1-year follow-up period. These results suggest that Hole ICL implantation is a suitable treatment also for low-vault myopic eyes.

Further, we believe that this paper will be of interest to the readership of your journal because this anterior segment surgery requires no peripheral iridectomy, as for conventional ICL. Moreover, conventional ICL implantation in low-vault eyes increases the risk for cataract formation and rotation of toric ICLs. In the present study, no such postoperative complications were observed after 1 year. This advances the field of refractive surgery.

Thank you for your consideration. I look forward to hearing from you.

---

## [Editor Report · Decision Letter 1]

21 Oct 2020

Assessment of low-vault cases with an implantable collamer lens

PONE-D-20-24129R1

Dear Dr. Kato,

We’re pleased to inform you that your manuscript has been judged scientifically suitable for publication and will be formally accepted for publication once it meets all outstanding technical requirements.

Kind regards,

Hidenaga Kobashi, M.D., Ph.D.

Academic Editor

PLOS ONE

Additional Editor Comments (optional):

The reviewers and editor have completed their assessments of your manuscript

Assessment of low-vault cases with an implantable collamer lens (PONE-D-20-24129R1)

and would like to publish it in the journal.
---

## [Editor Report · Acceptance letter]

26 Oct 2020

PONE-D-20-24129R1 

Assessment of low-vault cases with an implantable collamer lens 

Dear Dr. Kato:

I'm pleased to inform you that your manuscript has been deemed suitable for publication in PLOS ONE. Congratulations! Your manuscript is now with our production department. 

Kind regards, 

on behalf of

Dr. Hidenaga Kobashi 

Academic Editor

PLOS ONE